# Improvement of Tomato Fruit Quality and Soil Nutrients through Foliar Spraying Fulvic Acid under Stress of Copper and Cadmium

Xiaodi Shi, Lingfei Zhang, Zehan Li, Xiangyang Xiao, Nanbiao Zhan and Xiumin Cui *

College of Resources and Environment, National Engineering Laboratory for Efficient Utilization of Soil and Fertilizer Resources, Shandong Agricultural University, Tai'an 271018, China
* Correspondence: xiumincui@sdau.edu.cn

**Abstract:** Fulvic acid (FA), the essence and most active component in humus, is widely used as a fertilizer synergistic agent and for soil improvement. As a synergist, FA can not only highly chelate microelements, but also play a key role as a growth promoter. Due to the small molecular weight and high solubility, FA is usually used by foliar spraying in vegetable production, yet the effect on fruit quality and nutrient absorption is still unclear. Here, 'Jinpengdashuai' tomatoes were used to investigate the effects of spraying FA on tomato fruit quality and soil Cu and Cd availability under stress of Cu and Cd by pot experiments. The results showed that the 1000 mg·L$^{-1}$ FA could significantly improve the biomass of tomato plants to some extents under different stresses of Cu and Cd. After spraying FA, the Cu and Cd content in different organs and the whole accumulation decreased; meanwhile, the transport efficiency of Cu and Cd was also reduced to some degree. The drops of FA significantly changed the chemical form of Cu and Cd in 0~10 cm soil, from the easily absorbed water soluble (or ion) form into the insoluble form, hard to absorb. The Cu content in the residual state increased by 93.8% and 172.5%, respectively, under single and compound stress, and the Cd content in the residual state increased by 16.7% and 58.6%. Foliar spraying FA could promote the absorption and transport of essential nutrients such as nitrogen, phosphorus, potassium, calcium, magnesium and zinc, and maintain the nutrient balance, which alleviates the inhibition of normal metabolism by Cu and Cd stress, to a certain extent. More distinctly, Vc, total sugar and lycopene increased by 11.4~45.9%, 19.2~48.5% and 30.9~84.5%, respectively, indicating that foliar spraying FA could improve the appearance and flavor quality of tomato fruits under stress of Cu and Cd.

**Keywords:** copper stress; cadmium stress; tomato plant; fruit quality; mineral element

## 1. Introduction

Fulvic acid is an organic compound with the smallest molecular weight of humic acid. It contains a variety of active functional groups and has strong biological activity. It is soluble in both acidic and alkaline solutions [1], and is easily absorbed by plants. The structure of FA contains a large number of active groups such as carboxyl, phenolic hydroxyl, alcohol hydroxyl and carbonyl [2], which gives it the chelation ability for many kinds of heavy metals [3] and makes it form strong complexation and adsorption to heavy metal ions, thus affecting the morphology and distribution of heavy metals in soil and water [4]. FA, which could govern the environmental geochemistry behavior of heavy metals, is considered as an eco-friendly substance for controlling heavy metal pollutants in the environment, and the effect of FA on heavy metals in the environment depends on the type and state of heavy metals in the environment, and the state of heavy metals in the environment determines the biological activity and toxicity of heavy metals [5]. Research shows that FA produced positive effects on the quality of lemon fruits and could be used in lemon planting to improve the quality and added value of lemons [6]. FA could significantly increase chlorophyll content, leaf photosynthetic rate, and nitrogen use

efficiency, stimulate plant growth and increase the yield of maize [7]. FA could promote seed germination and root, stem and leaf growth of spring wheat and other crops, and improve the quality and yield of agricultural products [8]. FA could improve the vegetative characteristics, yield and nutrient uptake capacity of rice [9].

Tomato (Lycopersicon esculentum Mill. L) is one of the largest vegetables in both open field and facility cultivation mode. There are serious problems of high multi-cropping index and frequent fertilization during tomato production. Meanwhile, with the application of organic fertilizer, chemical fertilizer, pesticide and insecticide, more and more heavy metals are brought into the vegetable field, among which Cu and Cd are the most common pollution sources. FA is usually used as a trace element synergist or plant growth regulator for small molecular weight, and is usually applied to the foliar fertilization in facility vegetables production.

Currently, research on the effects of topdressing FA on the absorption and transport of mineral nutrients and fruit quality under heavy metal stress is unclear.

Here, this research will exploit the effects of foliar-spraying FA on tomatoes under Cu and Cd stress in a solar greenhouse, in order to provide some theoretical and practice guidance for the safe production of vegetables in areas with moderate and mild heavy metal pollution.

## 2. Materials and Methods

### 2.1. Test Material

Pot experiments were performed in the greenhouse of the College of Resources and Environment, Shandong Agricultural University (Longitude: 117.117673 E, Latitude: 36.19451 N) from 2020 to 2021. The tomato variety (Lycopersicon esculentum Mill.) was 'Jinpengdashuai'. The pot experiment was conducted as a randomized complete block design (RCBD). The suitable concentrations of FA, $Cu^{2+}$ and $Cd^{2+}$ were determined by preliminary experiments. FA (effective content 79%) was derived from QuanlinJiayou Fertilizer Company (Liaocheng, China). It contained N 5.796 $mg\cdot g^{-1}$, P 0.185 $mg\cdot g^{-1}$, K 1.593 $mg\cdot g^{-1}$, and was prepared in an aqueous solution with a concentration of 1000 $mg\cdot L^{-1}$ by distilled water. Tested soil was brown soil, the pH 7.72, and the electrical conductivity (EC) was 0.174 $Ms\cdot cm^{-1}$. The physical and chemical properties of the tested soil are displayed in Table 1.

**Table 1.** Physical and chemical properties of the tested soil.

| Indicators | The Tested Soil |
|---|---|
| Organic matter/($g\cdot kg^{-1}$) | 13.41 |
| $NO_3^-$/($mg\cdot kg^{-1}$) | 3.35 |
| $NH_4^+$/($mg\cdot kg^{-1}$) | 31.35 |
| Available P/($mg\cdot kg^{-1}$) | 40.06 |
| Available K/($mg\cdot kg^{-1}$) | 133.3 |
| Total Cu/($mg\cdot kg^{-1}$) | 8.54 |
| Total Cd/($mg\cdot kg^{-1}$) | 0.624 |
| Available Cu/($mg\cdot kg^{-1}$) | 1.23 |
| Available Cd/($mg\cdot kg^{-1}$) | 0.051 |

### 2.2. Test Method and Process

A total of 7 treatments were set up with 8 pots for each treatment, randomly arranged (Table 2). There was a total of 17.0 kg soil per pot (the upper inner diameter: 32.5 cm, the lower inner diameter: 25.5 cm, the height: 27.5 cm).

**Table 2.** The experimental treatments.

| Treatment Code | Treatment | Abbreviation |
|----------------|-----------|--------------|
| CK | Control | CK |
| T1 | Single Cu stress | Cu |
| T2 | Foliar spraying FA under Cu stress | Cu + FA |
| T3 | Single Cd stress | Cd |
| T4 | Foliar spraying FA under Cu stress | Cd + FA |
| T5 | Combined stress of Cu and Cd | Cu + Cd |
| T6 | Foliar spraying FA under Cu, Cd stress | Cu + Cd + FA |

Soil pretreatment: air-dried, mixed, sieved (40 mesh). By adding exogenous Cu and Cd, the total amount of Cu and Cd in the soil reached 150 mg·kg$^{-1}$ and 8 mg·kg$^{-1}$, respectively (Cu and Cd were provided by $CuCl_2 \cdot 2H_2O$ and $CdCl_2 \cdot 2.5H_2O$, respectively). Preparation of 200 μmol·L$^{-1}$ solution of $Cu^{2+}$ and $Cd^{2+}$, stored in dark at 4 °C, diluted according to the required concentration, evenly sprayed on the soil, added water to 80% of the field capacity, 2 weeks of pre-culturing. The contents of available Cu and available Cd in soil before planting were 24.70 mg·kg$^{-1}$ and 2.50 mg·kg$^{-1}$, respectively.

Plant pretreatment: the plants with the same growth vigor were selected for transplanting at the tomato seedling stage (3–4 true leaves), 1 plant per pot. When the seedlings grew the first inflorescence, foliar spraying FA 100 mL, equal amount of water for CK (distilled water was used in this experiment), 5 times in total. Measuring plant height, stem diameter and chlorophyll every 10 days. Urea 10.318 g, superphosphate 19.471 g and potassium sulfate 10.3972 g were applied three times in each pot, and then uniformly applied to soil after dissolving in water.

Fruits were harvested after natural ripening, yield recorded. When most of the fruits of the second and third spikes matured, two fruits were selected from each plant, washed with deionized water, cut into small pieces. Some of the fresh samples were used for fruit quality determination, and the remaining samples were placed in glassware, deactivated at 105 °C for 30 min, dried at 75 °C, ground, sieved, and sealed to be tested. At the same time, two pots of complete plants were collected from each treatment, the roots, stems and leaves were separated, washed, deactivated at 105 °C for 30 min, and dried to constant weight at 70 °C, and the dry biomass calculated.

The 0~10 cm surface soil samples in the pot were collected, cleaned, air-dried, sieved and sealed to be tested.

### 2.3. Measuring Projects and Methods

2.3.1. Determination of Different Forms of Cu and Cd in Soil

A total of 5.00 g air-dried soil sample was placed in a 50 mL plastic centrifuge tube.

Water-soluble state: 25 mL deionized water was added to the above centrifuge tube, shaken for 2 h, centrifuged for 20 min, and the supernatant was taken as the water-soluble test solution. The soil sample was cleaned with a small amount of deionized water, centrifuged, and the supernatant discarded.

Exchangeable state: 25 mL 1 mol·L$^{-1}$ $MgCl_2$ solution was added to the filter residue, shaken at 25 °C for 2 h, and centrifuged for 15 min. The soil sample was cleaned with a small amount of deionized water, centrifuged, and the supernatant discarded.

Carbonate state: At the previous step, 25 mL 1 mol·L$^{-1}$ NaAc solution was added to the filter residue, pH 5.0, shaken at 25 °C for 5 h, and centrifuged for 15 min. The soil sample was cleaned with a small amount of deionized water, centrifuged, the supernatant discarded.

Fe-Mn binding state: 25 mL of 0.04 mol·L$^{-1}$ $NH_2OH \cdot HCl$ was added to the filter residue in the previous step, shaken for 3 h, and centrifuged for 15 min. The soil sample was cleaned with a small amount of water and centrifuged to discard the supernatant.

Organic binding state: 5 mL 0.02 mol·L$^{-1}$ $HNO_3$ and 5 mL 30% $H_2O_2$ were added to the filter residue in the previous step, and the mixture was shaken at 85 °C for 2 h, and then 5 mL 30% $H_2O_2$ was added (repeat the above steps until the organic matter completely

reacted). Then, the mixture was shaken at 85 °C for 3 h, cooled down to room temperature, and 10.0 mL mixture of 3.2 mol·$L^{-1}$ $NH_4Ac$ and 20% $HNO_3$ added, shaken for 30 min, filtered. The supernatant was the test solution of the organic binding state.

Residue state: The residue left by the previous filtration was washed with deionized water, then dried at 100 °C, ground, and bagged for use. A total of 0.2000 g of the dried residual sample was weighed in a polytetrafluoroethylene crucible and extracted with $HCl$-$HNO_3$-$HClO_4$-$HF$ for preparation.

### 2.3.2. Determination of Cu, Cd and Trace Elements in Tomato Plants

The Cu, Cd, Fe, Mn, Zn, Ca and Mg content of tomato plants were determined by Atomic Absorption Spectrophotometry (AA-7000) after digesting by $HNO_3$-$HClO_4$.

### 2.3.3. Determination of Tomato Fruit Quality Index

Lycopene was extracted with methanol-toluene and determined by spectrophotometry [10]; sodium hydroxide titration of organic acids [11]; determination of Vc by oxalic acid-EDTA extraction method [12]; HCL boiling water bath, colorimetric determination of total sugar and reducing sugar [13].

### 2.4. Data Processing

Office 2017 and Origin (Origin Lab, Northampton, Massachusetts, USA) were performed for data processing and plotting. The least significant difference (LSD) was used for multiple comparisons between different treatment means.

## 3. Results and Analysis

### 3.1. Effects of Spraying FA on the Growth Vigor of Tomato under Different Cu, Cd Stress

Seen from Figure 1A, before using FA (May 4th), the plant height of the tomato seedlings treated with T1 (Cu) and T5 (Cu + Cd) decreased significantly, 7.94% and 9.12% lower than CK, respectively. On the 10th day (May 14th), compared with CK, the effect of stress enhanced the plant height of the tomato seedlings treated with T1 (Cu), T3 (Cd) and T5 (Cu + Cd), which significantly decreased by 16.1%, 13.6% and 16.1%, respectively, while in T4 (Cd + FA) and T6 (Cu + Cd + FA) treatments, spraying FA increased by 6.27% and 5.20% compared with stress conditions, significantly. On the 20th day after spraying FA (May 24th), the plant height of the tomato seedlings treated with T1 (Cu), T3 (Cd) and T5 (Cu + Cd) was significantly lower than CK, decreased by 11.6%, 11.6% and 12.2%, respectively. In T2 (Cu + FA), T4 (Cd + FA), T6 (Cu + Cd + FA) treatments, spraying FA significantly alleviated 4.05%, 5.21%, and 5.56% compared with stress conditions, respectively.

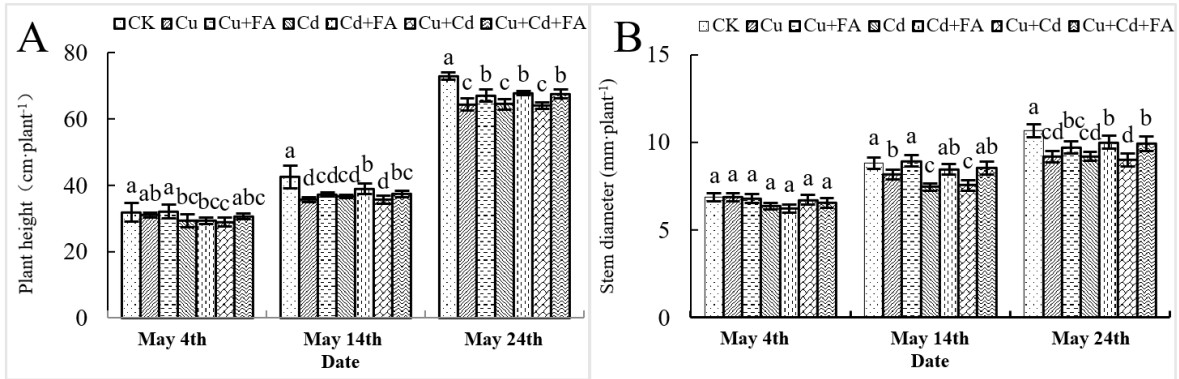

**Figure 1.** Plant height (**A**) and stem diameter (**B**) of tomato plants under different stresses of Cu, Cd treatments. The same alphabet above each column shows no significance ($p > 0.05$), in contrast, having significance ($p < 0.05$).

With the treatment time prolonged, the alleviating effect of FA on stress gradually became prominent. Before spraying FA (May 4th), there was no significant difference in the tomato stem diameter among the treatments, which might be due to the short cultivation time, and the stress effect of Cu, Cd was not significant (Figure 1B). On the 10th day after FA treatment (May 14th), Cu, Cd stress significantly inhibited the growth. The stem diameter of tomatoes treated with T1 (Cu), T3 (Cd) and T5 (Cu + Cd) was 7.17%, 15.4% and 14.3% lower than that of CK, respectively, while in T2 (Cu + FA), T4 (Cd + FA) and T6 (Cu + Cd + FA) treatments, spraying FA increased by 9.29%, 13.6% and 13.1% compared with stress conditions, respectively. On the 20th day after treatment (May 24th), the treatment with T1 (Cu), T3 (Cd) and T5 (Cu + Cd) had a stronger inhibitory effect on the tomato stem diameter, which was 13.7%, 13.8% and 15.4% lower than CK, respectively, while in the treatment T2 (Cu + FA), T4 (Cd + FA) and T6 (Cu + Cd + FA), spraying FA increased by 5.55%, 8.84% and 10.3% compared with stress conditions, respectively.

### 3.2. Effect of Spraying FA on Tomato Yield under Different Cu, Cd Stress

Compared with CK, the tomato yield of T1 (Cu), T3 (Cd) and T5 (Cu + Cd) decreased by 9.97%, 11.6%, and 23.7%, respectively, and significantly (Figure 2). After spraying FA, the treatment with T2 (Cu + FA), T4 (Cd + FA) and T6 (Cu + Cd + FA) increased by 7.41%, 7.40%, and 11.6% compared with stress conditions, respectively, which was significant, but did not completely return to the CK level.

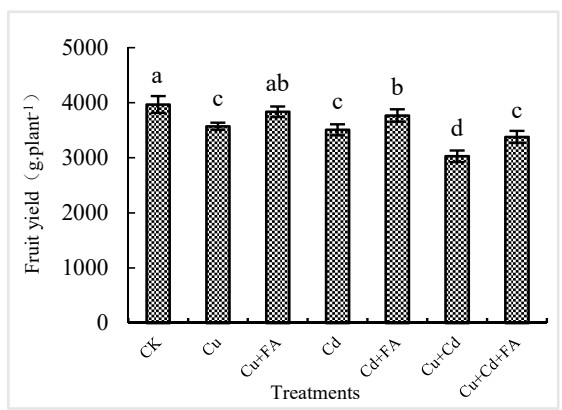

**Figure 2.** Spraying FA promoted fruit yield of tomato under different Cu, Cd treatments. The same alphabet above each column shows no significance ($p > 0.05$), in contrast, having significance ($p < 0.05$).

### 3.3. Effects of Spraying FA on Cu and Cd Contents in Tomato Fruit under Different Cu, Cd Stress

The application of FA significantly reduced the content of Cu and Cd in tomato fruit under stress. Compared with CK, the Cu content of tomato fruit treated with T1 (Cu) and T5 (Cu + Cd) increased significantly, by 41.1% and 360%, respectively (Figure 3A). In T5 (Cu + Cd) treatment, the addition of Cd promoted the absorption and transport of Cu in tomato under Cu stress. In T2 (Cu + FA) and T6 (Cu + Cd + FA) treatments, spraying FA significantly reduced the Cu content of tomato fruit, with a decrease of 13.1% and 57.7%, compared with stress conditions, respectively.

In CK, T1 (Cu) and T2 (Cu + FA) treatments, Cd was not detected in tomato fruits, while the Cd contents in tomato fruits treated with T3 (Cd) and T5 (Cu + Cd) were 0.0834 mg·kg$^{-1}$ and 0.0775 mg·kg$^{-1}$, respectively (Figure 3B). Adding Cu under Cd stress could reduce the absorption and transport of Cd in tomato. Compared with stress conditions, spraying FA decreased the Cd content significantly by 34.9% and 48.5%, in T4 (Cd + FA) and T6 (Cu + Cd + FA) treatments.

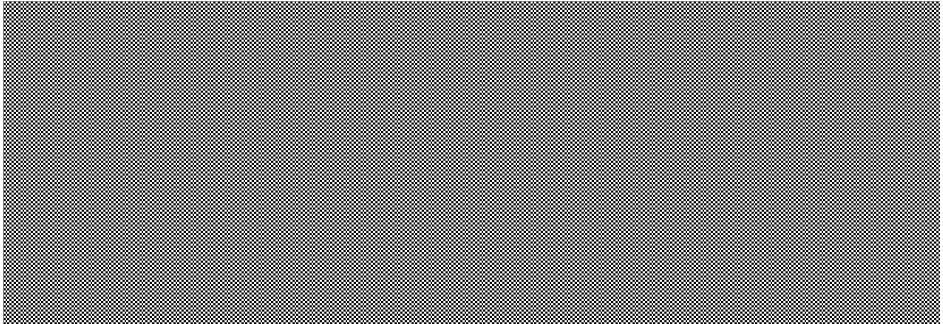

**Figure 3.** Effects of spraying FA on Cu, Cd in tomato fruit under different Cu (**A**), Cd (**B**) stress. The same alphabet above each column shows no significance (*p* > 0.05), in contrast, having significance (*p* < 0.05).

### 3.4. Effects of Spraying FA on the Fruit Quality of Tomato under Cu, Cd Stress

The Vc content of tomato fruit treated with T1 (Cu), T3 (Cd) and T5 (Cu + Cd) decreased significantly by 33.7%, 29.7% and 30.7%, compared with CK, respectively (Figure 4A). In T2 (Cu + FA), T4 (Cd + FA) and T6 (Cu + Cd + FA) treatments, spraying FA significantly alleviated the inhibition of Cu, Cd stress on the synthesis of Vc in tomato fruit compared with stress conditions; Vc increased by 45.9%, 26.9% and 11.4%, respectively, and the relief effect under Cd stress was the largest, the Vc content even returned to CK level.

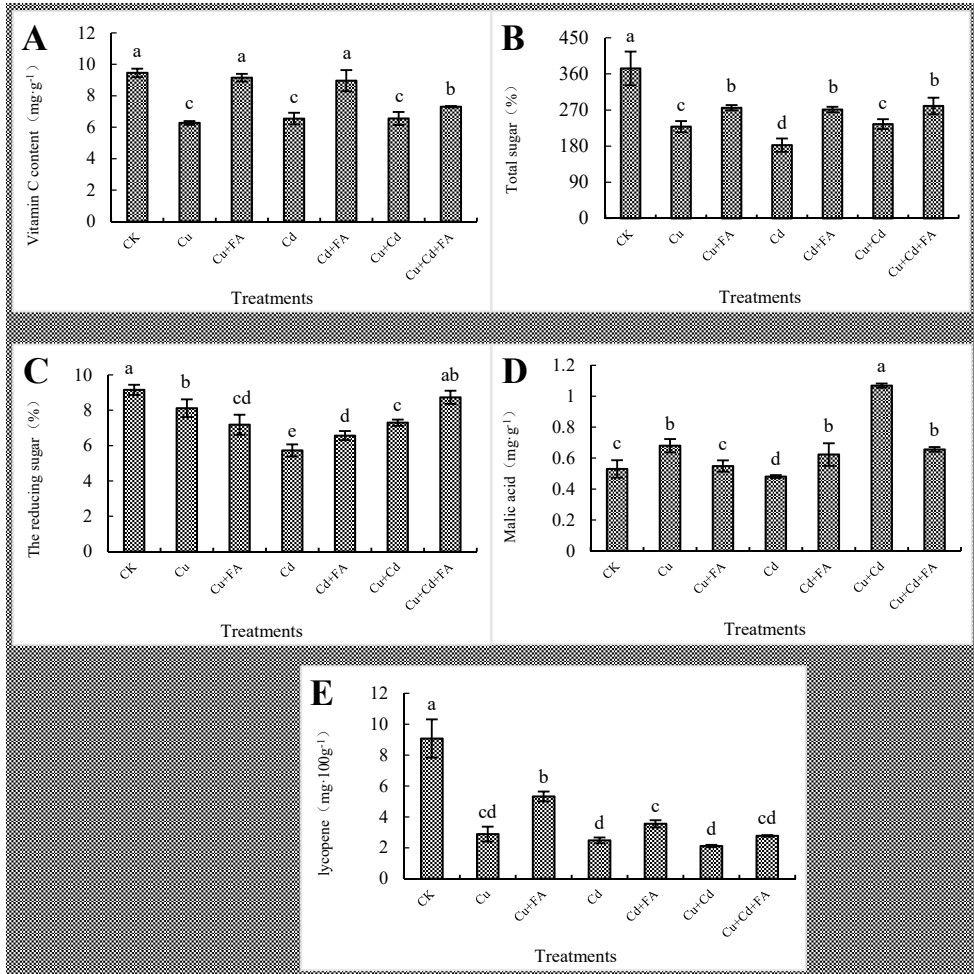

**Figure 4.** The content of Vc (**A**), total sugar (**B**), reducing sugar (**C**), malic acid (**D**) and lycopene (**E**) of tomato fruit under different Cu, Cd stress. The same alphabet above each column shows no significance (*p* > 0.05), in contrast, having significance (*p* < 0.05).

The total sugar content of tomato fruits treated with T1 (Cu), T3 (Cd) and T5 (Cu + Cd) decreased significantly by 38.8%, 51.1% and 37.1% compared with CK, respectively (Figure 4B). In T2 (Cu + FA), T4 (Cd + FA) and T6 (Cu + Cd + FA) treatments, spraying FA significantly alleviated the inhibition of Cu, Cd stress on the total sugar of tomato fruit compared with stress conditions, and the total sugar content increased by 20.6%, 48.5% and 19.2%, respectively, but did not return to CK level.

The reducing sugar content of tomato fruit treated with T1 (Cu), T3 (Cd) and T5 (Cu + Cd) decreased significantly by 11.4%, 37.4% and 20.3% compared with CK, respectively (Figure 4C). In T4 (Cd + FA) and T6 (Cu + Cd + FA) treatments, spraying FA significantly alleviated the inhibition of reducing sugar in tomato fruit compared with stress conditions, and the content of reducing sugar increased by 14.6% and 19.7%, respectively. However, in T2 (Cu + FA) treatment, spraying FA reduced the reducing sugar content of the tomato fruit by 11.5%.

The malic acid content of tomato fruits treated with T1 (Cu) and T5 (Cu + Cd) increased significantly, by 28.2% and 101.5%, respectively, while T3 (Cd) treatment reduced the malic acid content of tomato fruits by 9.35% compared with CK (Figure 4D). In T2 (Cu + FA) and T6 (Cu + Cd + FA), spraying FA significantly alleviated the inhibition of malic acid in tomato fruit compared with stress conditions, and the content of malic acid decreased by 19.3% and 38.6%, respectively, while in T4 (Cd + FA) spraying FA increased by 29.6% compared with stress conditions.

The lycopene content of tomato fruits treated with T1 (Cu) and T5 (Cu + Cd) decreased significantly by 68.2%, 72.7% and 76.7% compared with CK, respectively (Figure 4E). In T2 (Cu + FA), T4 (Cd + FA) and T6 (Cu + Cd + FA) treatment, spraying FA significantly alleviated the inhibition of stress on lycopene compared with stress conditions, which increased by 84.5%, 43.3% and 30.9%, respectively, but did not return to CK level.

### 3.5. Effects of Spraying FA on N, P, and K Content of Tomato Fruit under Different Cu, Cd Stress

Compared with CK, the N content of tomato fruit treated with T1 (Cu), T3 (Cd) and T5 (Cu + Cd) decreased significantly by 20.5%, 21.7% and 27.3%, respectively (Figure 5A). In T2 (Cu + FA), T4 (Cd + FA) and T6 (Cu + Cd + FA) treatment, spraying FA significantly alleviated the inhibition of stress on N in tomato fruit compared with stress conditions, and the N content was restored by 30.4%, 23.0% and 45.5%, respectively. T2 (Cu + FA) and T6 (Cu + Cd + FA) even exceeded CK.

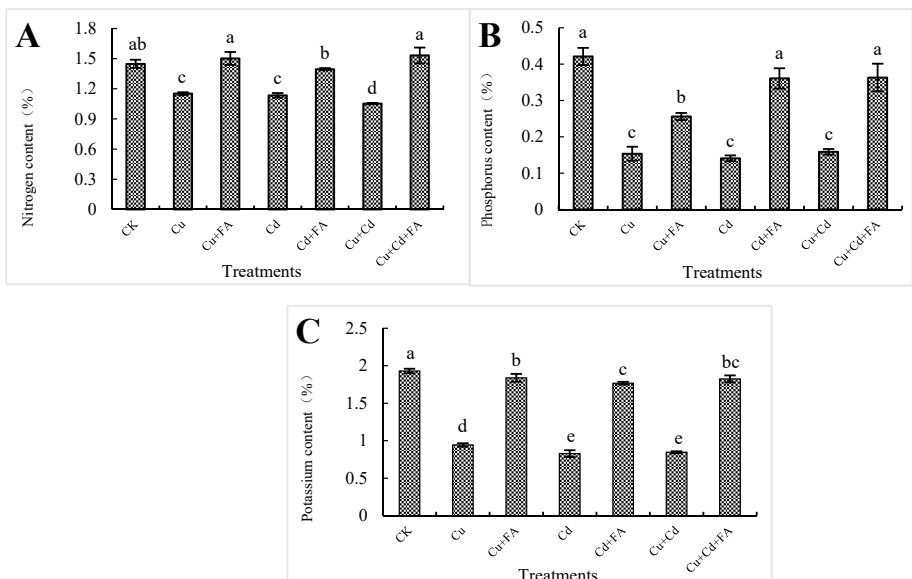

**Figure 5.** Alleviating Effects of FA on N (**A**), P (**B**), and K (**C**) of tomato fruit under different Cu, Cd stress. The same alphabet above each column shows no significance ($p > 0.05$), in contrast, having significance ($p < 0.05$).

Compared with CK, the P content of tomato fruit treated with T1 (Cu), T3 (Cd) and T5 (Cu + Cd) decreased significantly by 63.6%, 66.6% and 62.3%, respectively (Figure 5B). In T2 (Cu + FA), T4 (Cd + FA) and T6 (Cu + Cd + FA) treatment, spraying FA significantly alleviated the inhibition of stress on the P content of tomato fruit compared with stress conditions, the P content returned by 66.8%, 156% and 129%, respectively, and all returned to or exceeded CK.

Compared with CK, the K content of tomato fruit treated with T1 (Cu), T3 (Cd) and T5 (Cu + Cd) decreased significantly by 51.1%, 57.1% and 56.2%, respectively (Figure 5C). In T2 (Cu + FA), T4 (Cd + FA) and T6 (Cu + Cd + FA) treatment, spraying FA significantly alleviated the inhibition of stress on K absorption of tomato fruit compared with stress conditions, and the K content increased by 95.1%, 114% and 116%, respectively, but did not return to CK level.

### 3.6. Effects of Spraying FA on Fe, Zn, Ca and Mg in Tomato Fruit under Cu, Cd Stress

The contents of Fe, Zn, Ca and Mg in the tomato fruits were significantly different among treatments (Table 3). In T1 (Cu), T3 (Cd) and T5 (Cu + Cd) treatments, compared with CK, the content of Fe in the tomato fruits decreased significantly by 44.5%, 35.8% and 53.0%, respectively; the content of Zn decreased by 37.9%, 18.2% and 35.6%, respectively; the content of Ca decreased by 20.5%, 21.2% and 23.2%, respectively; the content of Mg decreased significantly by 7.35%, 6.95% and 8.87%, respectively.

**Table 3.** The content of Fe, Zn, Ca and Mg of tomato fruits under different Cu, Cd stress (mg·kg$^{-1}$).

| Element | Treatments | | | | | | |
|---------|------------|-----------|--------------|-----------|--------------|--------------|-------------------|
| | CK | T1 (Cu) | T2 (Cu + FA) | T3 (Cd) | T4 (Cd + FA) | T5 (Cu + Cd) | T6 (Cu + Cd + FA) |
| Fe | 210.5 ± 28.4 a | 116.9 ± 7.03 cd | 166.6 ± 7.33 b | 135.3 ± 4.64 c | 173.5 ± 18.8 b | 98.98 ± 5.17 d | 127.5 ± 5.55 c |
| Zn | 22.54 ± 2.99 a | 14.00 ± 0.74 c | 22.24 ± 1.08 a | 18.44 ± 0.62 b | 22.34 ± 1.70 a | 14.51 ± 0.28 c | 16.40 ± 1.07 bc |
| Ca | 0.16 ± 0.01 a | 0.12 ± 0.01 cd | 0.14 ± 0.00 b | 0.12 ± 0.00 cd | 0.13 ± 0.00 b | 0.12 ± 0.00 d | 0.13 ± 0.00 bc |
| Mg | 0.13 ± 0.00 a | 0.12 ± 0.00 b | 0.13 ± 0.00 a | 0.12 ± 0.00 b | 0.13 ± 0.00 a | 0.11 ± 0.00 b | 0.12 ± 0.00 a |

The same alphabet on the right side of the same list shows no significance ($p > 0.05$), in contrast, having significance ($p < 0.05$).

In T2 (Cu + FA), T4 (Cd + FA) and T6 (Cu + Cd + FA) treatment, compared with stress conditions, spraying FA increased the content of Fe by 42.5%, 28.3% and 28.8%, respectively, but did not return to CK; it increased the content of Zn by 95%, 114% and 116%, respectively, and some treatments returned to CK; it increased the content of Ca by 12.9%, 9.08% and 10.1%, respectively, but did not return to CK; and it increased the content of Mg by 8.20%, 7.65% and 6.67%, respectively, and all returned to CK. Spraying FA alleviated the inhibition of stress on the accumulation of Fe, Zn, Ca and Mg in tomato fruits.

### 3.7. Effects of Spraying FA on Cu and Cd Speciation in Soil under Cu, Cd Stress

For surface soil of 0~10 cm, the content of various forms of Cu increased significantly in T1 (Cu) and T5 (Cu + Cd) treatments (Table 4). Compared with stress conditions, in T2 (Cu + FA) treatment, spraying FA reduced the water-soluble, ion-exchanged, carbonate-bound, Fe-Mn oxide-bound, and organic-bound fractions by 15.5%, 37.7%, 26.3%, 51.2%, and 43.0%, respectively, and increased the residual fraction by 93.8%; in T5 (Cu + Cd) treatment, spraying FA reduced the water-soluble state, ion exchange state, carbonic acid bound state and Fe-Mn oxide-bound state of Cu by 21.1%, 30.5%, 12.8% and 27.6%, respectively, and increased the organic bound state and residual state by 16.1% and 172.5%, respectively. This shows that, under stress conditions, spraying FA promoted the transformation of Cu in the surface soil from an easily absorbed state to a not easily absorbed state.

**Table 4.** The contents of different Cu forms in topsoil under different Cu, Cd treatments (mg·kg$^{-1}$).

| Treatment | Water Soluble State | Ion Exchange State | Carbonate Combination State | Ferric–Manganese Oxidation State | Organic State | Residue State | Total |
|---|---|---|---|---|---|---|---|
| CK | 0.08 ± 0.01 d | 0.41 ± 0.06 c | 0.27 ± 0.01 d | 0.07 ± 0.01 d | 0.73 ± 0.03 d | 0.51 ± 0.12 d | 2.07 d |
| T1 (Cu) | 5.61 ± 0.52 a | 5.02 ± 0.41 a | 4.37 ± 0.55 a | 1.64 ± 0.05 a | 4.79 ± 0.18 a | 4.86 ± 1.14 b | 16.3 c |
| T2 (Cu + FA) | 4.74 ± 0.05 b | 3.13 ± 0.20 b | 3.22 ± 0.05 b | 0.80 ± 0.12 c | 2.73 ± 0.03 c | 9.42 ± 0.45 a | 24.0 a |
| T3 (Cd) | 0.47 ± 0.01 c | 0.45 ± 0.04 c | 0.55 ± 0.02 d | 0.09 ± 0.02 d | 0.28 ± 0.00 d | 0.19 ± 0.05 d | 2.03 d |
| T4 (Cd + FA) | 0.36 ± 0.02 c | 0.34 ± 0.02 c | 0.47 ± 0.01 d | 0.07 ± 0.00 d | 0.17 ± 0.01 d | 0.28 ± 0.08 d | 1.69 e |
| T5 (Cu + Cd) | 6.08 ± 0.49 a | 5.05 ± 0.69 a | 1.25 ± 0.03 c | 0.98 ± 0.02 b | 3.71 ± 0.94 b | 3.23 ± 0.11 c | 20.3 b |
| T6 (Cu + Cd + FA) | 4.80 ± 0.59 b | 3.51 ± 0.23 b | 1.09 ± 0.03 c | 0.71 ± 0.05 c | 4.42 ± 0.03 a | 8.80 ± 0.58 a | 23.3 a |

The same alphabet on the right side of the same list shows no significance ($p > 0.05$), in contrast, having significance ($p < 0.05$).

For the surface soil, the content of Cd in different forms was significantly different under different Cu, Cd stress treatments (Table 5). After the tomato harvest, no water-soluble Cd was detected in the soil of each treatment, and no ion exchange Cd was detected in the treatment without Cd addition. Spraying FA significantly reduced the exchangeable Cd in the surface soil and increased the proportion of residual Cd. In T3 (Cd) and T5 (Cu + Cd) treatments, the detected forms of Cd were higher than those of CK. Compared with stress conditions, in T4 (Cd + FA) treatment, spraying FA reduced the exchangeable and carbonate-bound Cd by 40.2% and 66.7%, respectively, while the organic-bound and residual Cd increased by 12.9% and 58.6%, respectively; in T6 (Cu + Cd + FA) treatment, spraying FA reduced the contents of ion-exchanged, carbonate-bound and organic-bound Cd by 38.1%, 9.09% and 18.9%, respectively, while the contents of Fe-Mn oxidation and residual Cd increased by 5.26% and 16.7%, respectively. On the whole, the foliar application of FA had the most significant effect on the exchangeable Cd and carbonate Cd, with large solubility in the surface soil, and the promotion of the Fe-Mn oxidation state, organic binding state and residual state with decreasing solubility showed an increasing trend, so it significantly reduced the bioavailability of Cd.

**Table 5.** The contents of different Cd forms in the topsoil under different Cu, Cd stress (mg·kg$^{-1}$).

| Treatment | Water Soluble State | Ion Exchange State | Carbonate Combination State | Ferric–Manganese Oxidation State | Organic State | Residue State | Total |
|---|---|---|---|---|---|---|---|
| CK | — | — | 0.20 ± 0.01 b | 0.06 ± 0.00 d | 0.02 ± 0.01 b | 0.07 ± 0.01 c | 0.35 b |
| T1 (Cu) | — | — | 0.14 ± 0.03 b | 0.06 ± 0.00 d | 0.05 ± 0.00 b | 0.17 ± 0.01 c | 0.44 b |
| T2 (Cu + FA) | — | — | 0.01 ± 0.00 c | 0.15 ± 0.01 c | 0.08 ± 0.01 b | 0.11 ± 0.02 c | 0.35 b |
| T3 (Cd) | — | 0.87 ± 0.10 a | 0.51 ± 0.09 a | 0.32 ± 0.04 b | 0.31 ± 0.03 a | 0.58 ± 0.20 b | 2.59 a |
| T4 (Cd + FA) | — | 0.52 ± 0.11 b | 0.17 ± 0.06 b | 0.32 ± 0.05 b | 0.35 ± 0.09 a | 0.97 ± 0.25 a | 2.33 a |
| T5 (Cu + Cd) | — | 0.97 ± 0.08 a | 0.22 ± 0.08 b | 0.38 ± 0.07 ab | 0.37 ± 0.10 a | 0.54 ± 0.18 b | 2.48 a |
| T6 (Cu + Cd + FA) | — | 0.60 ± 0.06 b | 0.20 ± 0.05 b | 0.40 ± 0.04 a | 0.37 ± 0.02 b | 0.73 ± 0.27 ab | 2.30 a |

The same alphabet on the right side of the same list shows no significance ($p > 0.05$), in contrast, having significance ($p < 0.05$).

## 4. Discussion

### 4.1. Effects of Spraying FA on Contents of Different Forms of Cu, Cd in Soil under Stress of Cu, Cd

Many forms of heavy metals are found in the soil, and their mobility and toxicity are different [14]. Heavy metal morphology is divided into the exchangeable state, oxidizable state, reducible state, and residue state [15]. It is generally said that the exchangeable state is the most harmful to the soil ecosystem and is easy to absorb by animals and plants [16]. FA contains abundant oxygen-containing functional groups (phenolic hydroxyl, carbonyl, carboxyl, etc.), which can promote the complexation with metal oxides and heavy metals [17]. It is a ubiquitous organic matter in the environment, and can enhance the

mobility and bioavailability of $Cd^{2+}$ through competitive complexation to form FA-heavy metal ions (FA-HMIs) complexes with excellent solubility [18]. In comparison to the above-ground portions, the root showed a higher bioaccumulation ability of Cd and Cu [19]. Here, for 0~10 cm surface soil, spraying FA could promote the transition of Cu in surface soil from an easily absorbed state to a more difficult absorbed state under stress conditions. Spraying FA could significantly reduce the exchangeable Cd in the surface soil and increase the proportion of residual Cd. Similar effects have been found in lettuce grown with cadmium toxicity [20]. Thus, it is conjectured that exogenous FA could reduce the speciation and bioavailability of Cu and Cd in topsoil.

*4.2. Effects of Spraying FA on the Growth and Mineral Nutrients of Tomatoes under Stress of Cu, Cd*

In this study, the plant height and stem diameter of the tomatoes decreased significantly under Cu, Cd stress, indicating that the initial soil available Cu (24.7 mg·kg$^{-1}$) and Cd (2.35 mg·kg$^{-1}$) inhibited the growth of the tomato, with Cd having a higher inhibiting effect than Cu [21], but still maintaining the basic growth. After spraying FA, the growth of the tomato plants recovered significantly, especially under Cd stress where the recovery of tomato growth was highest, suggesting that the spraying of FA enhanced the tolerance of tomato plants to Cu and Cd stress. From the absorption and accumulation of nitrogen, phosphorus, potassium, calcium, magnesium, iron, manganese and zinc in tomato plants, after spraying FA, the absorption of N, P and K in tomato fruits was significantly increased (20.5~156%) under stress of Cu, Cd, even more than CK. Studies showed that $Ca^{2+}$, $Mg^{2+}$ and other cations alleviated $Cu^{2+}$ and $Cd^{2+}$ toxicity to the roots through competition for the adsorption sites on the roots [22]. From the accumulation of Fe, Zn, Ca and Mg in tomato fruit, the effect of Cu and Cd stress on Fe and Zn was significantly higher than that of Ca and Mg, and the mitigation effect of spraying FA on tomato plants increased accordingly. This might be due to the competitive inhibition of Zn on the absorption of Cu and Cd by plants [23]. Studies showed that FA had the ability to improve nutrient uptake of crops [24]. Therefore, it was speculated that FA was involved in the regulation of nutrient transport in tomato plants, and spraying FA could promote the absorption of various mineral nutrients by tomato plants. Studies showed that heavy metals (such as Cu and Cd) had toxic effects on crops [25], and root activity is significantly reduced under high concentrations of $Cu^{2+}$ [26], thus affecting crop nutrient absorption and growth. Elevated levels of Cd in the soil were known to negatively affect the growth and fruit quality of the tomato [27]. Exogenous FA could not only effectively improve soil available nutrients [28], but also alleviate the inhibition of excessive $Cu^{2+}$ on enzyme activity [29]. Exceeding values of Cd in crops might cause serious human health risks [30]. Here, the Cu content of tomato fruits was significantly increased under single Cu and Cu + Cd combined stress. The addition of Cd promoted the absorption and transport of Cu by tomatoes under Cu stress, and spraying FA significantly reduced the Cu content of tomato fruit. The addition of Cu could reduce the absorption and transport of Cd in tomatoes under Cd stress, and the Cd content in tomato fruit was significantly reduced after spraying FA. This might be due to the excellent adsorption performance of FA on heavy metal cations ($Cu^{2+}$, $Cd^{2+}$) through the complexation and surface precipitation [31]. It indicated that spraying FA (1000 mg·kg$^{-1}$) could alleviate the transport of metal elements Cu and Cd to fruit, especially under Cd stress, and the mitigation effect of FA was the most significant. It was speculated that the application of FA could effectively reduce the Cu, Cd content of tomato fruit under Cu, Cd stress.

Studies showed that spraying FA promoted crop growth and improved fruit quality [32]. Similarly, in this study, the contents of Vc, total sugar, reducing sugar, malic acid, lycopene and other substances in tomato fruits decreased under Cu, Cd and Cu + Cd combined stress, and the decrease was significantly alleviated after spraying FA, especially the contents of Vc and total sugar, which were significantly higher than those under stress. It was speculated that spraying FA could improve tomato fruit quality under stress of Cu,

Cd. Tomato fruit yield decreased significantly under Cu, Cd and Cu + Cd combined stress. After spraying FA, tomato yield increased under each stress treatment. The alleviation of FA on the tomato fruit yield under Cu, Cd stress might be due to the improvement of FA on root growth, thereby increasing nutrient absorption [33], and the greater the decline in tomato yield, the stronger the mitigation effect of FA. It was speculated that spraying FA could increase tomato yield under the stress of Cu, Cd.

## 5. Conclusions

(1) Foliar spraying of FA changed the forms of Cu and Cd in 0~10 cm soil, made them transform to insoluble and residual forms, reduced the bioavailability of Cu and Cd, reduced the absorption and transport of Cu and Cd in tomato plants, and further reduced the accumulation of heavy metals in fruits.

(2) Foliar spraying FA promoted the absorption of various mineral nutrients under stress of Cu, Cd, especially increased the absorption of N, P and K by tomato fruits (20.5~156%), improved the quality of tomato fruits, especially the content of Vc (11.4~45.9%) and total sugar (19.2~48.5%), and increased the yield of tomato fruits.

(3) Under stress of Cu, Cd, foliar spraying FA could exert multiple effects, including promoting nutrient absorption and reducing the bioavailability of heavy metals in soil and their accumulation in fruits. Therefore, the desired goals of improving quality, increasing yield, and reducing pollution could be achieved synchronously.

**Author Contributions:** Conceptualization and methodology, X.C. and X.S.; software, X.S.; validation and formal analysis, X.S. and L.Z.; investigation and resources, X.S. and L.Z.; data curation, X.S. and Z.L.; writing—original draft preparation, L.Z. and Z.L.; writing—review and editing, X.X. and N.Z.; visualization, supervision, project administration, and funding acquisition, X.C. All authors have read and agreed to the published version of the manuscript.

**Funding:** This research was funded by the Natural Science Foundation of Shandong Province (ZR2021MC145) and the Key Research and Development Program of Shandong Province (2021CXGC010801).

**Data Availability Statement:** Shandong Agricultural University.

**Conflicts of Interest:** The authors declare no conflict of interest.

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
