# Peer review of "Improvement of Tomato Fruit Quality and Soil Nutrients through Foliar Spraying Fulvic Acid under Stress of Copper and Cadmium"

_agronomy, doi:10.3390/agronomy13010275_

Round 1
Reviewer 1 Report
The manuscript entitled “Spraying fulvic acid improved tomato fruit quality and activated soil nutrients under copper and cadmium stress'', studies that the effects of foliar spraying fulvic acid on tomato under copper, cadmium stress in solar greenhouse, in order to provide some theoretical and practice guidance for safe production of vegetables in areas with moderate and mild heavy metal pollution. The authors presented that foliar-spraying with fulvic acid could improve the appearance and flavor quality of tomato fruit under copper or cadmium stress, increasing vitamin C, total sugar and lycopene.
My suggestions:
- The authors need to revise the title of the paper in a more meaningful way;
- The abstract is badly written. Please improve the abstract to cover the important topics reviewed and discussed in this article. The abstract is written in a way lacks logic. It should highlight the salient findings more critically;
- Keywords are present in the title, choose others;
- The introduction should be improved by highlighting the published studies related to this topic. Furthermore, the hypothesis and objectives of this review should be mentioned clearly in the introduction section. Introduction need more convincing rational for this article;
- Provide experimental work plan at the start of M&M. No detail description is available about the experimental design. What statistical method is used?
- Authors should discuss the results integrally. The discussion is based on individual results. I suggest that integrating the results will give more value to the work. I suggest that you discuss by integrating all your results. You can use correlation tests (PCA or Pearson Correlation).
- In Figures and tables, what does it mean to have the bars in the columns? Standard error or standard deviation? please specify and improve figure captions.
- The discussion is poorly written hence, needs rewriting. The discussion should be further strengthened by adding some more relevant papers. The literature search is insufficient, only few related research papers in the past three years are cited, add the latest research results appropriately. See the below links if you think it will benefit your discussion.
The discussion is poorly written hence, needs rewriting. The discussion should be further strengthened by adding some more relevant papers. The literature search is insufficient, only few related research papers in the past three years are cited, add the latest research results appropriately. See the below links if you think it will benefit your discussion. Organic matter, in addition to acting as an attenuator of the adverse condition, in most cases working as an abiotic tolerance signaling and elicitor. Canellas, L.P.; Olivares, F.L.; Aguiar, N.O.; Jones, D.L.; Nebbioso, A.; Mazzei, P.; Piccolo, A. Humic and fulvic acids as biostimulants in horticulture. Sci. Hortic. 2015, 196, 15–27. At the beginning of the discussion, better explore the possibilities of using transcriptomics microorganisms, Read and quote: Terra, L.A., de Soares, C.P., Meneses, C.H.S.G. et al. Transcriptome and proteome profiles of the diazotroph Nitrospirillum amazonense strain CBAmC in response to the sugarcane apoplast fluid. Plant Soil 451, 145–168 (2020). https://doi.org/10.1007/s11104-019-04201-y
- The conclusion is confused. Rewrite the conclusion! It needs to be much improved.
Author Response
Reply to the reviewer 1
TITLE:
The authors need to revise the title of the paper in a more meaningful way.
Reply: The title has been changed to " Improvement of Tomato Fruit Quality and Soil Nutrients Through Foliar Spraying Fulvic Acid under Stress of Copper and Cadmium”.
ABSTRACT:
The abstract is badly written. Please improve the abstract to cover the important topics reviewed and discussed in this article. The abstract is written in a way lacks logic. It should highlight the salient findings more critically.
Reply: The abstract has been modified.
KEY WORDS:
Keywords are present in the title, choose others.
Reply: Modified, deleted “fulvic acid”, added “tomato plant”, see line 27.
INTRODUCTION:
The introduction should be improved by highlighting the published studies related to this topic. Furthermore, the hypothesis and objectives of this review should be mentioned clearly in the introduction section. Introduction need more convincing rational for this article.
Reply: Modified, added related papers,​cleared the hypothesis and objectives, see line 57-62.
Provide experimental work plan at the start of M&M. No detail description is available about the experimental design. What statistical method is used?
Reply: Added” randomized complete block design (RCBD)”, see line 68. Added table 1 and table 2 to explain the experimental soil and detailed treatments. See line 73-74 and line 78-79
RESULTS:
Authors should discuss the results integrally. The discussion is based on individual results. I suggest that integrating the results will give more value to the work. I suggest that you discuss by integrating all your results. You can use correlation tests (PCA or Pearson Correlation).
Reply: Modified.
In Figures and tables, what does it mean to have the bars in the columns? Standard error or standard deviation? please specify and improve figure captions.
Reply: Modified, see line 273.
DISCUSSION:
The discussion is poorly written hence, needs rewriting. The discussion should be further strengthened by adding some more relevant papers. The literature search is insufficient, only few related research papers in the past three years are cited, add the latest research results appropriately.
Reply: Rewritten, added related research papers.
CONCLUSION:
The conclusion is confused. Rewrite the conclusion! It needs to be much improved.
Reply: Rewritten, see line 368-382.

Reviewer 2 Report
Although presented work has many interesting outcomes, it has been poorly written. It needs proofreading and rearrangement of the sections. There is a lack of referencing and discussion is not supported by literature at a sufficient level. Please see additional comments below.
Line62-65 Please put it as a table, will be more clear.
Line 68 How do you justify it? Is there any result confirming it? Otherwise you can just remove description about root.
Line 69-71 What do you mean by biochemical preparation? please rephrase this paragprah
Section 1.2 needs to be reduced and simplified. Is a lot of poorly written information. Please rewrite this part.
What are those digits after FA (5.4, 5.12 etc.)? Please explain
In result part there is too much information, it is better to mention main achievements, since many of those data are already on the graphs and tables.
Tables do not contain essential descriptions, what are those numbers, what is their unit
Discussion section 3.1 is again explanation of results. Please support your results with references.
Line 303 is there any reference to confirm this sentence?
Author Response
Reply to the reviewer 2
Line62-65 Please put it as a table, will be more clear.
Reply: Modified, see line 74-75 Table 1.
Line 68 How do you justify it? Is there any result confirming it? Otherwise you can just remove description about root.
Reply: Modified, deleted the description about root.
Line 69-71 What do you mean by biochemical preparation? please rephrase this paragprah
Reply: Modified, see line 70-72
Section 1.2 needs to be reduced and simplified. Is a lot of poorly written information. Please rewrite this part.
Reply: Rewritten,see Table 2
What are those digits after FA (5.4, 5.12 etc.)? Please explain
Reply: Modified, changed "5.4" to "May 4th" etc.
In result part there is too much information, it is better to mention main achievements, since many of those data are already on the graphs and tables.
Reply: Modified.
Tables do not contain essential descriptions, what are those numbers, what is their unit.
Reply: Modified, added unit, see line 270,275,287.
Discussion section 3.1 is again explanation of results. Please support your results with references.
Reply: Modified, added some references.
Line 303 is there any reference to confirm this sentence?
Reply: Modified, deleted this sentence.

Round 2
Reviewer 2 Report
Please refer to the Table/Table numbers in the main text. They stand alone.
Author Response
Reply to the reviewer 2
Line62-65 Please put it as a table, will be more clear.
Reply: Modified, see line 74-75 Table 1.
Line 68 How do you justify it? Is there any result confirming it? Otherwise you can just remove description about root.
Reply: Modified, deleted the description about root.
Line 69-71 What do you mean by biochemical preparation? please rephrase this paragprah
Reply: Modified, see line 70-72.
Section 1.2 needs to be reduced and simplified. Is a lot of poorly written information. Please rewrite this part.
Reply: Rewritten,see Table 2.
What are those digits after FA (5.4, 5.12 etc.)? Please explain
Reply: Modified, changed "5.4" to "May 4th" etc.
In result part there is too much information, it is better to mention main achievements, since many of those data are already on the graphs and tables.
Reply: Modified.
Tables do not contain essential descriptions, what are those numbers, what is their unit.
Reply: Modified, added unit, see line 269,273,285.
Discussion section 3.1 is again explanation of results. Please support your results with references.
Reply: Modified, added some references.
Line 303 is there any reference to confirm this sentence?
Reply: Modified, deleted this sentence.
